# Solidification Experiment of Lithium-Slag and Fine-Tailings Based Geopolymers

**Bi-Bo Dai [1], Yi Zou [2],\*, Yan He [3], Ming Lan [3] and Qian Kang [1]**

[1] State Key Laboratory of Safety and Health for Metal Mines, Maanshan 243000, China
[2] Zijin Mining Group Co., Ltd., Xiamen 572000, China
[3] School of Resources & Environment and Safety Engineering, University of South China, Hengyang 421001, China
\* Correspondence: zou.yi@zijinmining.rs

**Abstract:** Based on the pressure of environmental protection, more and more scientific researchers are trying to reuse aluminum–silicon-rich industrial wastes. In this study, activated lithium-slag and lead–zinc tailings were used as raw materials to prepare geopolymers at ratios of 3:7, 1:1, and 7:3. These geopolymers were initially cured for 12 h at 25 °C, 50 °C, 75 °C, and 100 °C and were then cured at room temperature to the specified ages. The compressive strength of each group of geopolymers was tested at the ages of 3 days, 7 days, and 28 days. The optimal group of samples was selected, that is, those with a ratio of lithium-slag to lead–zinc tailings of 7:3 and an initial curing temperature of 75 °C. After that, the heavy metal leaching test and porosity analysis test were carried out on the optimal group of samples, and the curing effect was considered to meet the requirements of the Chinese specifications. In addition, in order to reveal the mechanism of the chemical reaction, scanning electron microscopy and X-ray diffraction (XRD) were used to study the microstructure and hydration products of the C3 group cured samples. This study provides a new concept for the reuse of industrial wastes such as lithium-slag and fine-tailings.

**Keywords:** sustainable mining; tailings reuse; unclassified tailings; lithium-slag; geopolymer; environmental protection





## 1. Introduction

The "reuse of industrial wastes" and ensuring that this process is "safe" are necessary for the sustainable development of the mining industry [1]. At present, the reuse of solid waste resources is a popular subject of research worldwide. Many researchers have used aluminum–silicon-rich solid waste to make geopolymer, which is a new type of green gelling material. Its main raw material can be aluminosilicate minerals occurring in nature, or it can be made from solid industrial waste with a high aluminum–silicon content [2–4]. Studies have shown that blast furnace slag [5], zeolite [6], metakaolin [7], fly ash [8], red mud [9], and other synthetic geopolymers have many advantages, such as high strength, low toxicity substance leaching rates, good durability, etc. Such synthetic geopolymers are widely used in industrial applications such as brick-making and road construction.

Advancement in the manufacturing industry has led to the development of the mining industry and improved the capacity of mineral recycling. Compared with the last century, the tailings produced by new mines are finer, higher in amounts of mud, and more difficult for dehydration, which is not conducive to building tailings dams [10]. According to analyses of the tailings from many metal mines around the world, the content of aluminum and silicon materials in the tailings from metal mines is relatively high, especially in the tailings from lead–zinc mines. In tailings from lead–zinc mines, the content of aluminum and silicon varies from 60% to 88%, the particle size is extremely fine, the specific surface area is large, and these tailings have potential gelling properties, enabling them to be used as raw materials to make geopolymers [11–13]. At the same time, as mining work

is becoming deeper and deeper, the pressure of surrounding rocks is becoming greater and greater, and the requirements for the pressure and shear ability of the past are also becoming higher and higher. Therefore, it is necessary to develop gel materials with better durability and higher strength to meet the needs of deep mining [14]. China is a major mining country, which produces a large number of tailings every year [15]. Many Chinese researchers conducted related research on the reuse of tailings, including the use of tailings for the preparation of geopolymer [16]. However, in the existing research, it is found that the silicon dioxide in the tailings is mainly quartz. This means that only using tailings to prepare geopolymer has an unsatisfactory effect. It is necessary to study adding solid wastes with high pozzolanic activity in a large proportion to improve the effect of gel condensation.

Lithium-slag is an industrial waste slag formed by smelting spodumene or lepidolite ore as a raw material to produce lithium products, and it is a type of non-ferrous metal smelting slag [17]. Lithium-slag glass structure is a polymeric network structure composed of a silicate polyhedron formed after roasting and smelting, and it belongs to the $SiO_2$-$Al_2O_3$-$CaO$ slag system [18]. Lithium-slag has pozzolanic activity and is a reusable industrial waste product that is widely used as a cement additive [19].

Based on the above status, in this study, lithium-slag and lead–zinc tailings containing high aluminum and silicon contents were used to make geopolymers, and these geopolymers were studied. The porosity, compressive strength, and leaching resistance of the solidified body were tested, and the microstructure of the solidified body was analyzed by means of scanning electron microscopy to explore the feasibility of preparing cementitious materials from fine-grained tailings and lithium slag.

## 2. Materials and Methods

### 2.1. Materials

Mixing silicon–aluminum raw materials with different chemical compositions and structural characteristics can realize the complementarity of the properties of silicon–aluminum raw materials, which is conducive to the adjustment and control of the polymerization reaction [20]. Therefore, in our experiments, fine-grained lead–zinc tailings that were rich in Si and Ca (taken from the exit of the tailing pumping station at the Yinshan lead–zinc mine in Jiangxi) and lithium-slag that was rich in Si and Al (taken from a carbonic acid mine in Yichun, Jiangxi) were selected for use as composite silicon–aluminum raw materials, and NaOH (purchased from Shenzhen Hongsen New Materials Technology Co., Ltd., Shenzhen, China, with a white sliced appearance, the solid content is greater than 99%) was selected as the pozzolanic active excitation material for lithium-slag (see Figure 1). The semi-quantitative element analysis and mineral composition analysis were carried out using an X-ray fluorescence spectrometer (XRF) and an X-ray diffractometer (XRD); the results are shown in Table 1 and Figure 2, respectively. The particle sizes for the lithium-slag and lead–zinc tailings were measured using a laser particle size analyzer. The average particle size for the lead–zinc tailings was only 54.69 μm, which corresponds to fine-grained tailings. The detailed results are shown in Table 2.

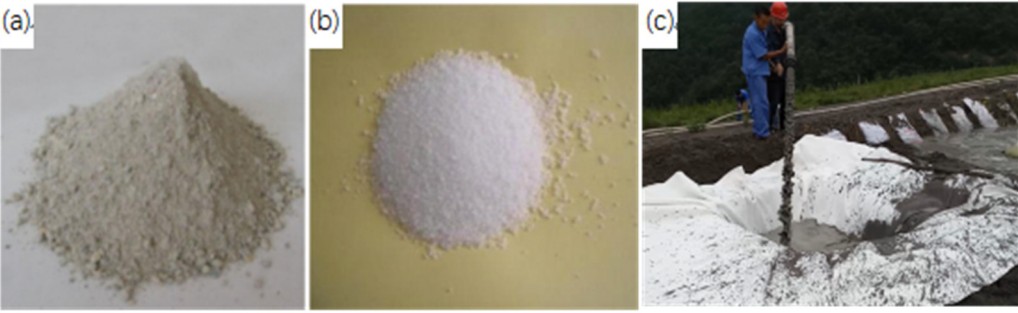

**Figure 1.** Materials: (**a**) lithium-slag; (**b**) flaky sodium hydroxide; (**c**) lead–zinc tailings.

**Table 1.** Physical and chemical properties of the fine-tailings and lithium-slag.

| | | Proportions (%) | | | | | | | | pH Value | Physical Properties | | |
|---|---|---|---|---|---|---|---|---|---|---|---|---|---|
| | Mineral | $SiO_2$ | $Al_2O_3$ | $Fe_2O_3$ | CaO | MgO | $CaSO_4$ | MnO | Others | | Specific Surface Area ($m^2$/kg) | Density (g/$cm^3$) |
| Sample | Fine-tailings | 49.36 | 16.25 | 9.11 | 3.32 | 3.30 | 3.24 | / | 15.42 | 7.8 | 267 | 2.71 |
| | Lithium-slag | 53.26 | 23.21 | 2.42 | 4.46 | 0.73 | 4.96 | 0.27 | 10.96 | 5.6 | 670 | 2.46 |

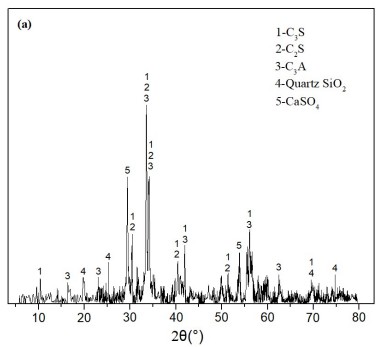 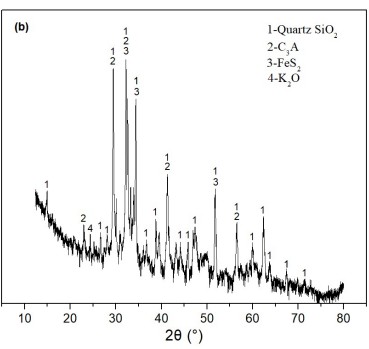

**Figure 2.** XRD patterns: (**a**) lithium-slag and (**b**) fine-tailings.

**Table 2.** Particle size composition of the fine-tailings and lithium-slag.

| | Unit/(μm) | ⁻d (bar) | $d_{10}$ | $d_{30}$ | $d_{50}$ | $d_{60}$ | $d_{80}$ | $C_u$ | $C_c$ |
|---|---|---|---|---|---|---|---|---|---|
| Sample | Fine-tailings | 54.69 | 7.57 | 24.87 | 53.91 | 71.96 | 121.89 | 9.51 | 0.67 |
| | Lithium-slag | 41.52 | 1.62 | 13.16 | 29.84 | 41.47 | 72.38 | 25.60 | 2.58 |

Note: ⁻d (bar) refers to the average diameter of the lithium-slag and fine-tailings; $C_u = d_{60}/d_{10}$; $C_c = (d_{30})/(d_{10} \times d_{60})$.

Figure 2 depicts the X-ray diffraction (XRD) test results for the lithium-slag and fine-tailings. As Figure 2 shows, the main components of the lithium-slag included quartz-$SiO_2$, tricalcium silicate ($C_3S$), dicalcium silicate ($C_2S$), tricalcium aluminate ($C_3A$), and calcium sulfate ($CaSO_4$). High concentrations of $Al_2O_3$ and $SiO_2$, which have potential gelling activity [21], were present in the lithium-slag. The fine-tailings were mainly formed of quartz-$SiO_2$, $C_3A$, pyrite ($FeS_2$), kaolinite, and clay.

The equipment used in the experiments included a muffle furnace, a high-performance vibration ball mill machine, an X-ray diffractometer (XRD) (Bruker D8 Advance, Karlsruhe, Germany), a scanning electron microscope (SEM) (JSM-6490LV Japan), a Winner 2308 Laser particle size analyzer (Jinan Winner Particle Instrument Stock Co., Ltd., Jinan, China), an RMT-150B rock mechanics test system (Wuhan Institute of Rock and Soil Mechanics, Wuhan, China), an Autopore IV 9510 automatic pore size distribution mercury porosimeter (Micromeritics Instrument Corporation, Atlanta, GA, USA), and an X Series II Inductively Coupled Plasma Mass Spectrometer (Thermo Fisher Scientific, Waltham, MA, USA).

### 2.2. Pretreatment of Raw Material

#### 2.2.1. Activation of Lithium-Slag

The lithium-slag used in this study was taken from a lithium carbonate smelting facility in Yichun, Jiangxi. This lithium-slag was solid industrial waste produced by smelting lithium carbonate products with spodumene as a raw material, and its chemical composition was relatively stable. The chemical composition of lithium-slag determines the network structure of its glass structure to a certain extent and has a strong influence on the gelling activity of lithium-slag [22]. Lithium-slag is mainly composed of a glass structure with high content of silicon and aluminum, and the content of $SiO_2$ and $Al_2O_3$

accounts for more than 70%. High temperature and alkali-induced recombination can accelerate the breakage of Si-O and Al-O bonds in lithium-slag glass structure, promoting the depolymerization of the glass structure and releasing more active substances, thereby increasing the activity of lithium-slag. Therefore, in this study, high-temperature chemical compound excitation was used to improve the potential pozzolanic activity of the lithium-slag [23].

According to the melting point of lithium-slag, the original lithium-slag, containing 10% NaOH after mixing, was calcined at a temperature of 1450 °C for 60 min, and nitrogen was used for protection during the heating process. After the calcination, the samples were cooled in cold water and the water-quenched lithium-slag samples were collected and dried in a vacuum for 6 h. A high-performance vibrating ball mill was used to vibrate and grind the water-quenched lithium-slag to obtain activated lithium-slag samples. The process is shown in Figure 3.

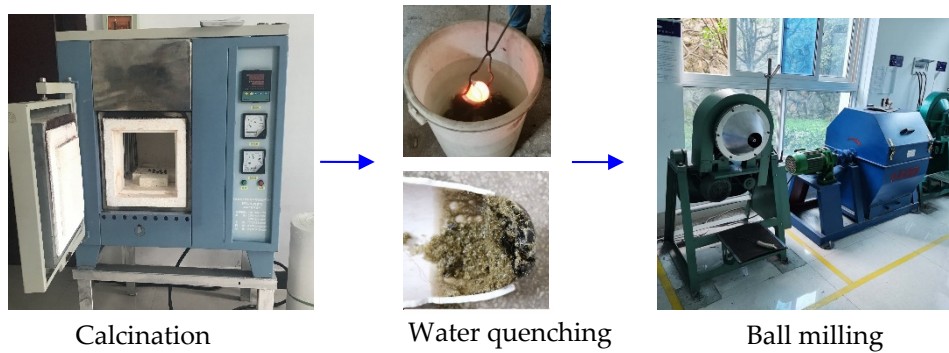

| Calcination | Water quenching | Ball milling |

**Figure 3.** Activation flow chart of lithium-slag.

The particle size distribution of the activated lithium-slag after grinding was measured by means of a laser particle size analyzer, and a particle size distribution table is presented in Table 3. It can be seen from the particle size grading results that particles smaller than 74 μm accounted for more than 80% of the activated lithium-slag after grinding, and particles smaller than 37 μm accounted for more than 60%. The content of fine particles was high, and the particle size distribution and continuity were very good.

**Table 3.** Particle size distribution composition of activated lithium-slag.

| particle size/μm | +250 | −250–+150 | −150–+75 | −75–+45 | −45–+37 | −37 |
|---|---|---|---|---|---|---|
| content/% | 1.10 | 2.76 | 10.49 | 10.24 | 8.87 | 66.54 |

In order to study the distribution characteristics of the crystalline phase and glass phase in activated lithium-slag, the sample was tested and analyzed via XRD, and the results are shown in Figure 4. A diffuse hump appeared in the lithium-slag sample between the 20° and 35° diffraction angles. Using HighScore software to analyze these results, we determined that the mineral phase represented by this diffuse hump was mullite. The mineralogical properties of related types of materials show that the diffuse hump corresponding to this interval is a typical feature of silicate glass. There was only a small amount of inconspicuous crystalline phase diffraction peaks in the XRD pattern of the sample, indicating that the activated lithium-slag under the condition of rapid cooling by water quenching had a high degree of vitrification. In addition to the diffuse hump that appeared in the range of diffraction angles of 20°–35°, a weak diffraction peak appeared in the sample near the position of 2θ = 40°, representing the mineral phase of magnetite [24].

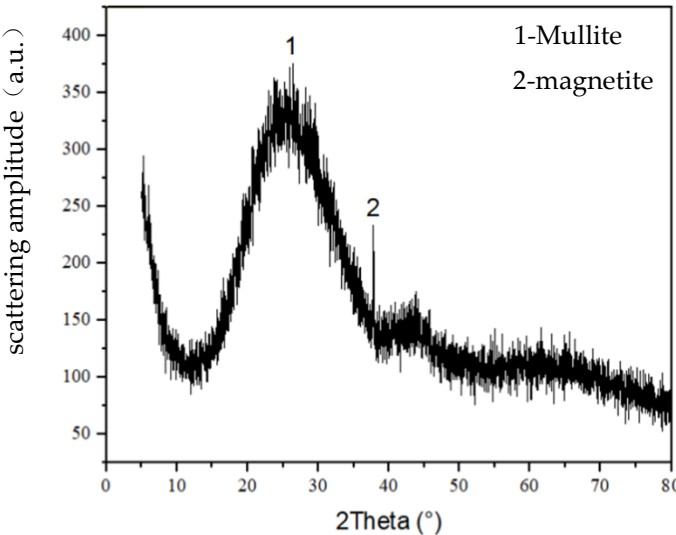

**Figure 4.** XRD patterns of the activated lithium-slag.

2.2.2. Activation of Lead–Zinc Tailings

The structure and properties of the crystalline quartz contained in lead–zinc tailings are very stable, but after activation via the addition of alkali and calcining, part of the $SiO_2$ in the form of quartz can be converted into active $SiO_2$. This pretreatment method activates the crystal lattice of the material and promotes the transformation of the high-polymerization state in the aluminum–silicon network structure into a low-polymerization state. The activated lead–zinc tailings can participate in the polymerization reaction and improve the curing effect [25].

In our experiments, we referred to previous research results [26]. In the first step, NaOH and tailings were weighed according to a mass ratio of 1/5; in the second step, solid NaOH was dissolved in a small amount of water to form a high-concentration alkaline solution (the amount was just enough to make NaOH completely soluble in water); the third step was to mix the alkali solution with the weighed lead–zinc tailings and put them in an oven to dry at 80 °C for 12 h; the fourth step was to place the sample obtained in the previous step into a muffle furnace and calcinate it at 500 °C for 1 h; in the fifth step, we ground the bulk calcined material obtained in the previous step for 15 min with a grinder and finally obtained activated pretreated tailings.

*2.3. Methods*

2.3.1. Experimental Design

An increase in the ratio of $Na_2O/SiO_2$ is beneficial to the formation of an aluminosilicate network structure, thus improving the strength of the geopolymer [27]. The proportion of $Na_2SiO_4$, as well as $SiO_2$, $Al_2O_3$, and $Na_2O$ in the mixture of lithium-slag and tailings is an important factor affecting the mechanical properties of geopolymers. $Na_2SiO_4$ used in this experiment was purchased from Tangshan Langshuo inorganic Silicone Co., Ltd. In this study, $Na_2SiO_4$ mixed with NaOH was used as the activator, NaOH and a small amount of water were added according to the $Na^+$ content in the liquid sodium silicate to adjust its modulus to 1.2, and it was packaged after completion.

First, we mixed the activated lithium-slag obtained from pretreatment with the lead–zinc tailings according to the mass ratios of 3:7, 1:1, and 7:3. Second, we added the activator and water according to the mass concentration of 64%, stirred the mixture evenly, and poured it into the model to make test cubes. Third, we put the test cubes obtained in the previous step into the oven at 25 °C, 50 °C, 75 °C, and 100 °C, and after curing for 12 h, demoulded them to make test specimens. A total of 12 groups of samples were made, and the details are shown in Table 4.

**Table 4.** Preparation conditions of specimen.

| Specimen Number | Lithium-Slag: Fine-Tailings | Curing Temperature/°C | Solid Proportions/% |
|---|---|---|---|
| A1 | 3:7 | 25 | 64 |
| A2 | 3:7 | 50 | 64 |
| A3 | 3:7 | 75 | 64 |
| A4 | 3:7 | 100 | 64 |
| B1 | 5:5 | 25 | 64 |
| B2 | 5:5 | 50 | 64 |
| B3 | 5:5 | 75 | 64 |
| B4 | 5:5 | 100 | 64 |
| C1 | 7:3 | 25 | 64 |
| C2 | 7:3 | 50 | 64 |
| C3 | 7:3 | 75 | 64 |
| C4 | 7:3 | 100 | 64 |

### 2.3.2. Experimental Process

Extensive experiments were carried out in this study to explore the feasibility of using activated lithium-slag and activated fine-tailings to make the geopolymer, as shown in Figure 5.

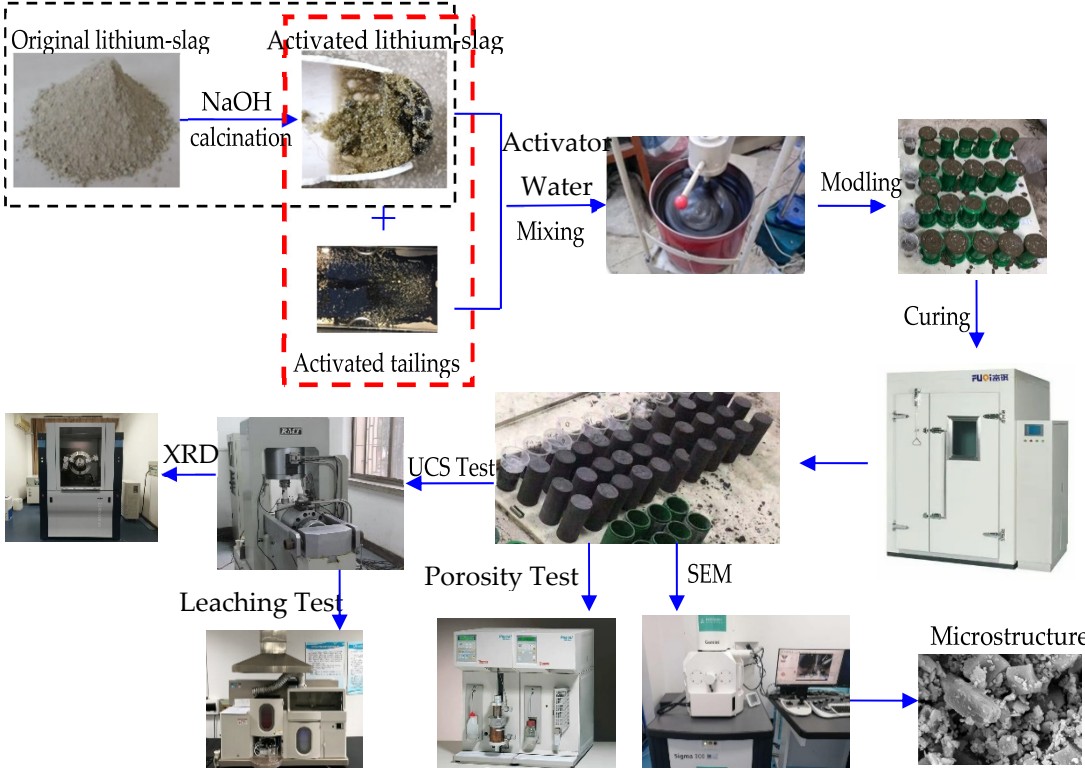

**Figure 5.** Schematic diagram of the experimental process.

(1)    Synthesis of Specimens

According to the conditions used for the preparation of specimens (Table 4), we first weighed the activated lithium-slag and the activated fine-particle-size tailings into the mixing tank according to the mass ratios 3:7, 1:1, and 7:3 and then added the configured activator and water until the mass concentration reached 64%. We stirred the specimens at 600 r/min for 3 min, poured the obtained slurry into the molds, and divided them into 12 groups according to the preparation conditions of each specimen (Table 4). We put them in an oven to cure for 12 h, and then cured them at room temperature for 28 days.

(2)　　Testing and Characterization

The compressive strengths of each group of samples were tested at 3 d, 7 d, and 28 d, and the effects of different curing temperatures and raw material ratios on the properties of the synthesized geopolymer were analyzed. Selecting a group of samples with the best tested effects, we tested their porosity and heavy metal leaching after the curing period (28 d) and used a scanning electron microscope (SEM) and an X-ray diffractometer (XRD) to test the microstructures of the geopolymers to analyze the mechanism of the solidification of heavy metals.

## 3. Results and Discussion

### 3.1. UCS

Uniaxial saturated compressive strength is the most important index used to measure the solidified bodies of tailings. In this experiment, the RMT-150B rock mechanics test system (Wuhan Institute of Rock and Soil Mechanics) was used to conduct three sets of parallel experiments on the test specimens, and the average value was taken as the experimental result under each condition. The 3 d, 7 d, and 28 d compressive strength results for samples in each group are shown in Table 5, the influence of the raw materials ratio on strength is shown in Figure 6, and the influence of the curing temperature on strength is shown in Figure 7.

**Table 5.** UCS performance of specimens.

| Specimen Number | UCS Test Results | | |
| --- | --- | --- | --- |
| | 3 d | 7 d | 28 d |
| A1 | 3.6 | 5.1 | 7.3 |
| A2 | 6.4 | 8.5 | 11.3 |
| A3 | 9.9 | 12.3 | 15.3 |
| A4 | 7.6 | 10.2 | 12.4 |
| B1 | 11.5 | 16.4 | 25.1 |
| B2 | 16.7 | 22.8 | 33.7 |
| B3 | 22.3 | 36.8 | 45.6 |
| B4 | 17.6 | 20.5 | 31.8 |
| C1 | 11.6 | 17.6 | 31.4 |
| C2 | 17.1 | 23.3 | 38.8 |
| C3 | 24.2 | 40.2 | 52.4 |
| C4 | 18.2 | 21.8 | 33.2 |

It can be seen from the comparison chart that increasing the content of lithium-slag greatly improved the curing effect. We speculated that the reason for this is that a large amount of glass structure is produced during the production of lithium-slag and water quenching, which has good pozzolanic activity and can participate in the polymerization reaction. Furthermore, the silicon in the tailings exists in the form of quartz, with good stability. Even if its activity is improved after activation, it cannot compare with the activity of the aluminum–silicon in the water-quenched lithium-slag.

With the increase in the curing temperature, the compressive strength of the sample first increased and then decreased. We speculated that the reason for this may be that with the increase in the curing temperature, the rate of depolymerization of aluminosilicate in the raw materials to form $[SiO_4]$ tetrahedron and $[AlO_4]$ tetrahedron was accelerated, and more gel substances were dissolved. At the same time, the number of micropores in the sample increased and the internal structure became denser, which finally increased the compressive strength of the sample. However, when the curing temperature exceeds a certain limit, the slurry will generate a large amount of gel in the initial stage of the reaction, which will cover the surfaces of unreacted raw materials and hinder the dissolution of unreacted silica and alumina. At the same time, a temperature that is too high will cause the free water in the solidified structure to evaporate extremely quickly, and the structure

of the geopolymer will crack, so the compressive strength of the sample will decrease with the increase in curing temperature at this time [28]. The optimum curing temperature for this test is 75 °C.

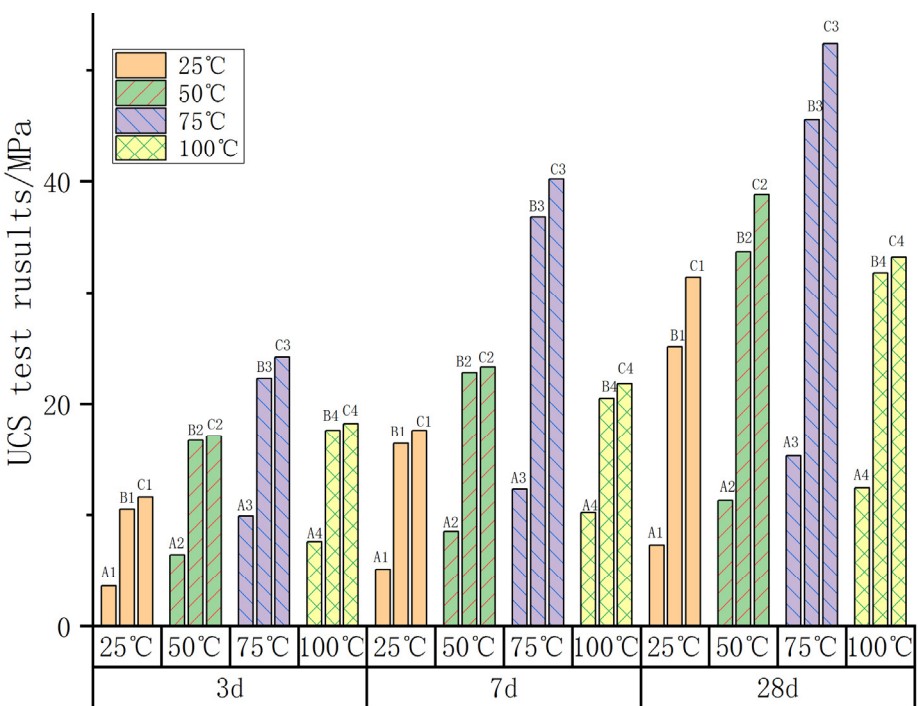

**Figure 6.** The effect of the raw materials ratio on the compressive strength of the specimens.

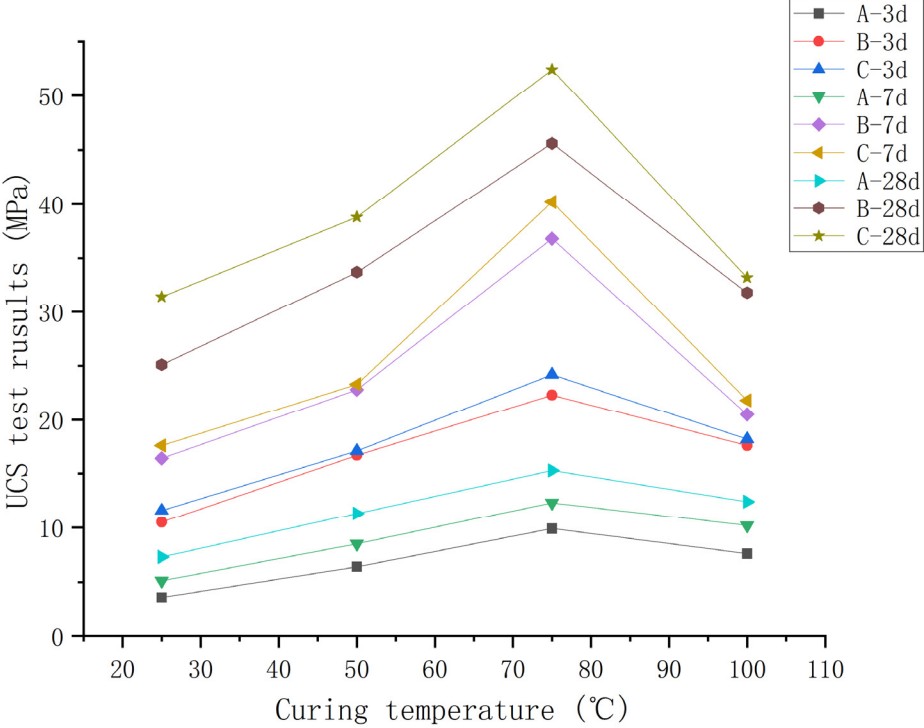

**Figure 7.** The effect of the curing temperature on the compressive strength of the specimens.

In this experiment, group C3 had the best curing effect, that is, the geopolymer prepared using a 7:3 ratio of lithium-slag to tailings and curing at 75 °C. The next step was

to conduct experiments on this group of specimens to test their heavy metal leaching rate and to conduct porosity and microstructure analyses to observe their curing mechanism.

### 3.2. Leaching Test

The leaching rate of solidified solids was closely related to the raw materials used in the preparation of the solidified solids and the curing method.

Lithium-slag itself contains a variety of trace heavy metal elements, such as manganese (Mn), chromium (Cr), cadmium (Cd), copper (Cu), etc. In addition, the lead–zinc ore mineral process results in lead–zinc tailings containing not only lead (Pb) and zinc (Zn) but also heavy metal elements such as arsenic (As) and barium (Ba). Therefore, in this experiment, the leaching experiment was carried out on the geopolymers from group C3, and the leaching concentrations of the above eight heavy metal elements were tested. According to the method for the determination of solid waste elements in the Identification Standards for Hazardous Wastes: Identification of Leaching Toxicity (GB 5085.1-2007), an X Series II Inductively Coupled Plasma Mass Spectrometer (ICP-MS) produced by Thermo Fisher Scientific was used for the heavy metal leaching tests. The experimental sample consisted of group C3 specimens with a curing age of 28 days, and the method was as follows:

(1) We weighed 50 g of the solid sample and placed it in a 500 mL extraction ampoule, added an appropriate amount of $H_2SO_4/HNO_3$ mixed liquid extractant (with a mass ratio of 3:1 and a pH value of $4.5 \pm 0.05$) according to a solid–liquid ratio of 1:14, sealed it, and fixed it in an overturned oscillatory device;

(2) We set the rotation speed to 30 r/min, the test temperature to $25\,°C \pm 1\,°C$, and the oscillation time to 16 h;

(3) After the vibration was completed, we removed the extraction bottle and let it stand for about 40 min. We installed a filter membrane soaked in dilute nitric acid in the suction filtration device, filtered the sample in the extraction bottle, and collected the filtrate;

(4) The heavy metal elements in the collected filtrate were detected via ICP-MS according to GB 5085.1-2007.

The heavy metal leaching results are shown in Table 6, and all results met the limit values required by Chinese regulations.

**Table 6.** Main heavy metal leaching concentration of specimen C3 (mg/L).

| Specimen | As | Cr | Cd | Pb | Zn | Cu | Mn | Ba |
|---|---|---|---|---|---|---|---|---|
| C3 | 0.706 | 0.002 | 0.001 | 0.088 | 0.154 | 0.104 | 0.001 | 0.094 |
| Specification limit | 0.500 | 0.200 | 0.040 | 1.000 | 5.000 | 2.000 | 0.100 | 25.000 |

### 3.3. Mechanism Analysis

3.3.1. SEM Analysis

We crushed the C3 group samples that had been destroyed in the rock mechanics experiments and took the core part. After drying and dehydration, using SEM, we scanned the sample with a very finely focused high-energy electron beam, thereby exciting the corresponding physical information for imaging. The SEM analysis results of a typical lithium-slag- and tailings-based polymer are shown in Figure 8.

It can be seen in the figure that the surface of the solidified body did not have obvious needle-like or flake-like features, but a very dense granular disorderly accumulation of connections. There was a small amount of pores on the surface of the cured body, which may have formed after the sample was destroyed, or may have been formed by air bubbles that were mixed in during curing. Granular gel products produced by the polymerization reaction could be observed in the fracture section. The gel changed the microstructure of the solidified body by covering, bonding, and filling and to a certain extent filled the pores between the incompletely activated tailings.

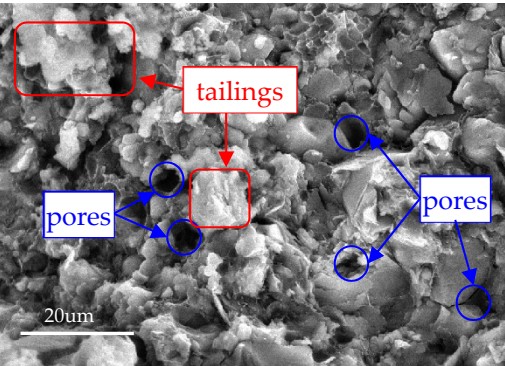

**Figure 8.** SEM micrographs of specimen C3 after 28 days of curing.

### 3.3.2. XRD Analysis

X-ray diffraction analysis was used to determine the phase composition of the sample. The equipment used in this experiment was a D8 Advance X-ray diffractometer, produced by Bruker, Germany. The transmitter power was 3 KW, and the sample was scanned in continuous scanning mode, with a scanning step width of 0.02°/step and a scanning speed of 5°/min (2θ). The absolute accuracy of the angle measurement was 0.01°, and the scanning range was 5°–90°.

It can be seen in the Figure 9 that the split diffraction feature peak of the geopolymer is long, the crystallization is not high, and the crystalline phase and the amorphous phase coexisted in the geopolymer sample. Four diffraction feature peaks appeared at 2θ = 21.2°, 27.8°, 54.2°, and 64.6°. The mineral characteristics of related types of materials are classified as the crystal phase of $SiO_2$, indicating that the feature peak of the quartz is still in geopolymer, but the strength of the diffraction peak has decreased significantly. In addition, two diffraction feature peaks appeared at 2θ = 32.8°, 35.6°, which are classified as the crystal phase of albite [29]. There was no alumina in the generated geopolymer, indicating that the alumina in the raw material participated in the polymerization reaction as a reactant. In addition, zeolite-like phases that were not present in the raw materials were found in geopolymers, indicating that the zeolite-like phases were new phases formed by reactions during material preparation.

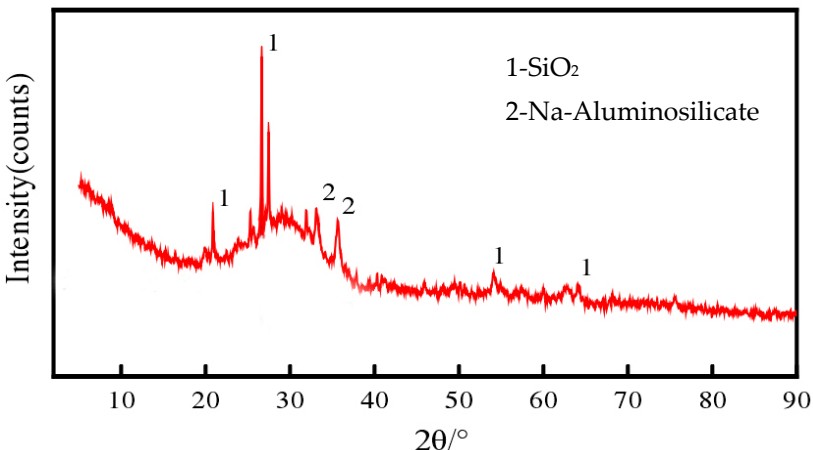

**Figure 9.** XRD patterns of C3 specimens after 28 days of curing.

### 3.3.3. Porosity Test

We tested the pore distribution of the lithium-slag-tailings-based geopolymer in the group C3 specimens. The equipment used in this experiment was an Autopore IV 9510 automatic pore-size distribution mercury porosimeter produced by Micromeritics in the United States. The applied mercury pressure was 0–414 MPa, and the measuring hole

radius range was 0.005–360 μm. First, the samples destroyed in the compressive strength tests were collected. Second, the samples were soaked in absolute ethanol and then dried in a vacuum at 50 °C for 6 h. Finally, the dried samples were used for the porosity tests. The tests showed that the total porosity of the sample was 23.46%, and the few harmful pores with pore diameters between 20 and 50 nm accounted for the majority. There was also a small number of harmful pores with pore diameters between 50 and 200 nm. The pore size distribution of C3 specimens after 28 days of curing is shown in Figure 10.

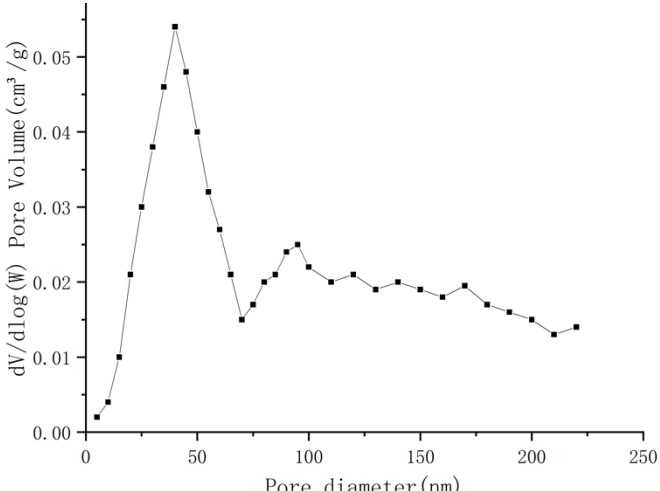

**Figure 10.** The pore size distribution of C3 specimens after 28 days of curing.

## 4. Conclusions

This study is the first time the feasibility of preparation of geopolymer using activated fine-tailings mix with lithium-slag was studied, which is of great significance for the reuse of fine tailings. In this study, geopolymers were prepared using different proportions of lithium-slag and tailings cured at different temperatures, and the optimal ratio and curing temperature combination was obtained according to the compressive strength of the solidified body. Leaching experiments were carried out on the optimal group of cured bodies to test the leaching effect of heavy metals. In order to obtain the solidification mechanism of the lithium-slag- and tailings-based polymer, the solidified body was tested for porosity and scanned using an electron microscope, and its microstructure was analyzed by means of an XRD. The main conclusions based on these experiments are as follows:

(1) After activation, lithium-slag and fine-grained tailings can be used to prepare geopolymers as cementitious materials, which represents a new concept for the reuse of industrial waste such as lithium-slag and tailings.

(2) The lithium-slag- and tailings-based polymer not only has a reasonable strength index but also has low porosity and few harmful pores. In addition, the obtained results regarding the leaching of heavy metals meet the requirements of the Chinese national specifications, indicating that this material can be further studied as a replacement for cement for the solidification of tailings.

(3) Future research should further study the influence of the proportion of lithium-slag and tailings on the solidified body, the influence of the activation process on the solidified body, the influence of the activator modulus on the solidified body, and the influence of the curing process on the solidified body to provide references for industrial applications.

**Author Contributions:** Conceptualization and investigation, B.-B.D.; methodology, Y.Z.; data curation, formal analysis, and validation, Y.H.; resources and supervision, M.L.; writing—original draft, Q.K. All authors have read and agreed to the published version of the manuscript.

**Funding:** This research was supported by the open research fund program of State Key Laboratory of Safety and Health for Metal Mines (Grant No. 2020-JSKSSYS-04).

**Institutional Review Board Statement:** Not applicable.

**Informed Consent Statement:** Not applicable.

**Data Availability Statement:** Not applicable.

**Conflicts of Interest:** The authors declare no conflict of interest.

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
