# Peer review of "Solidification Experiment of Lithium-Slag and Fine-Tailings Based Geopolymers"

_sustainability, doi:10.3390/su15054523_

Round 1
Reviewer 1 Report
The present manuscript entitled “Solidification experiment of lithium-slag and fine-tailings based geopolymers” by Dai et al., describes the activated lithium-slag and lead–zinc tailings that were used as raw materials to prepare geopolymers at different ratios. Furthermore, the compressive strength of each group of geopolymers was tested at the ages of 3, 7, and 28 days, and this present study provides the concept for the reuse of lithium slag and fine tailings. The authors report an interesting work. The objective and justification of the work are clear and I congratulate the authors for their good work. However, I recommend it for publication after certain Minor corrections are detailed below which need to be addressed before its final acceptance in Sustainability.
I advise the authors to take the following points into account while revising their manuscript.
Comment 1: There are some typographical errors in the manuscript text, so the authors need to correct them in the revised manuscript.
Comment 2: English needs to be a little improved, as there are some misused conjunctions and technical flaws to correct in the manuscript.
Comment 3: Revise the abstract section, it should clearly discuss the problem statement and the current study approach.
Comment 4: Since the introduction is too short, to improve the introduction section the authors need to cite and discuss some more recent references in the introduction section to strengthen the section.
Comment 5: please check and revise the keywords.
Comment 6: Please enhance the novelty statement at the end of the introduction section. Please add why the study is important and what are the outcomes of the study.
Comment 7: Include the structured graphical abstract in the revised manuscript to attain a broad readership.
Comment 8: Include the used Materials procured details in the revised manuscript.
Comment 9: In section 3.2.2., provide and discuss the XRD diffraction pattern peak positions of SiO2 and Na- Aluminosilicate in the manuscript text with some references.
Reviewer 2 Report
In this study, the geopolymers prepared by different proportions of lithium slag and tailings were cured at different temperatures, and the optimal ratio and curing temperature combination was obtained according to the compressive strength of the solidified body. The microstructure and hydration products of the optimal samples were also studied by the scanning electron microscope (SEM) and X-ray diffraction (XRD). The results provide a new idea for the reuse of lithium slag and fine tailings. There are some minor revisions for the further enhancement by adding some relevant references and discussions.
1. Please check the correctness of figure format and annotation carefully. For example, (α)(β)(χ) are presented in fig. 1.
2. Some relevant papers about preparing cementitious materials from fine-grained tailings and lithium slag should be cited, and the related discussions should be added to the manuscript.
3. In your introduction, please note and cite other, similar papers in this field and state why this paper is needed and how it is unique?
4. The English in the manuscript needs improvement. There were many errors in word form, use of articles ('a', 'an', 'the'), and use of correct number (singular vs plural).
5. In the text, some sentences are unreadable, and a lot of tense errors are found in this manuscript. Please carefully check the logicality of all sentences and make the article easy to read. In addition, the format of the citations should also be carefully checked.
6. In 2.2.1 section, authors analyzed and discussed the results shown in Figure 4. However, there is a lack of documentary evidence from related references.
Reviewer 3 Report
The paper described solidification experiment of lithium-slag and fine-tailings 2 based geopolymers. Overall the paper is well written and structured although some improved for language is needed. i would suggest acceptance with some language improvement. Please stress the innovation and contribution of the paper.
Reviewer 4 Report
Reuse of solid waste resources is a hot research topic currently. This paper tries to obtain a kind of geopolymers make up by different proportions of lithium slag and tailings. Then conducted leaching experiment, porosity experiment, scanning electron microscope experiment and XRD experiment on solidified body. The study presented in paper is innovative and interesting and has practical significance. This paper is publishable after some modification, the specified suggestions are as follows.
1.The activated lithium slag is mixed with the lead-zinc tailings according to mass ratio of 3:7, 1:1, and 7:3, the mass ratio is determined by what?
2. In the leaching test, how the parameters of overturned oscillatory device is determined?
3.In the process of specimens production, does it matter if the slurry is vibrated to reduce original bubbles in the sample after it is poured into the mold?
4. please figure out the basis of selecting the type of heavy metal elements that should be leached.
